# Comparative Study on the Sensing Kinetics of Carbon and Nitrogen Nutrients in Cancer Tissues and Normal Tissues Based Electrochemical Biosensors

**DOI:** 10.3390/molecules28031453

**Published:** 2023-02-02

**Authors:** Dingqiang Lu, Danyang Liu, Yujiao Liu, Xinqian Wang, Yixuan Liu, Shuai Yuan, Ruijuan Ren, Guangchang Pang

**Affiliations:** 1College of Biotechnology & Food Science, Tianjin University of Commerce, Tianjin 300134, China; 2Tianjin Key Laboratory of Food Biotechnology, Tianjin 300134, China; 3Tianjin Institute for Food Safety Inspection Technology, Tianjin 300134, China

**Keywords:** carbon and nitrogen nutrients, electrochemical biosensor, activation constant, nutrient sensing, signal amplification factor

## Abstract

In this study, an electrochemical sensor was developed by immobilizing colon cancer and the adjacent tissues (peripheral healthy tissues on both sides of the tumor) and was used to investigate the receptor sensing kinetics of glucose, sodium glutamate, disodium inosinate, and sodium lactate. The results showed that the electrical signal triggered by the ligand–receptor interaction presented hyperbolic kinetic characteristics similar to the interaction of an enzyme with its substrate. The results indicated that the activation constant values of the colon cancer tissue and adjacent tissues differed by two orders of magnitude for glucose and sodium glutamate and around one order of magnitude for disodium inosinate. The cancer tissues did not sense sodium lactate, whereas the adjacent tissues could sense sodium lactate. Compared with normal cells, cancer cells have significantly improved nutritional sensing ability, and the improvement of cancer cells’ sensing ability mainly depends on the cascade amplification of intracellular signals. However, unlike tumor-adjacent tissues, colon cancer cells lose the ability to sense lactate. This provides key evidence for the Warburg effect of cancer cells. The methods and results in this study are expected to provide a new way for cancer research, treatment, the screening of anticancer drugs, and clinical diagnoses.

## 1. Introduction

Cancer is one of the most difficult diseases to cure in the history of human diseases, and its incidence rate is increasing year by year [1]. Although the cause of the disease is highly complex [2,3], almost all cancers share a common significant metabolic marker, namely the Warburg effect, also known as aerobic glycolysis [4,5,6]. Healthy cells undergo glycolysis to produce lactic acid to provide energy in an oxygen-deficient environment. When oxygen is sufficient, this inefficient means of energy production is replaced by the tricarboxylic acid (TCA) cycle, whereas cancer cells continue to undergo glycolysis even with a sufficient oxygen supply. Numerous studies have shown that the reason that cancer cells undergo extensive aerobic glycolysis is mainly because their proliferation depends on their need for fatty acids, proteins, and nucleotides (DNA replication and transcription). The substances α-ketoglutarate and oxaloacetate in the TCA cycle are the precursors of glutamic acid and aspartic acid [7], respectively. Citric acid is derived from the glycolysis of pyruvate in the mitochondria (dehydrogenated and decarboxylated by the pyruvate dehydrogenase complex) and fatty acids (by β-oxidation), producing acetyl-CoA, and is shuttled into the cytoplasm as the raw material for plasma membrane synthesis [8,9]. In addition, the glycerophosphate shuttle or malate–aspartate shuttle is also the basic route for NADH + H^+^/NAD^+^ exchange between intra- and extramitochondrial compartments [10]. It is worth noting that the high-energy thioester bond of succinyl-CoA drives the phosphorylation of the GDP substrate to generate GTP, which is an important basis for GTP demand and its balance with ATP [11]. GTP is not only required for the initiation and elongation of protein translation but is also the basic phosphate carrier for G proteins (including small G proteins and heterotrimeric G proteins) in activation and signal amplification [12]. Among them, the small G protein is the driving force for the intra- and extranuclear information exchange and signal amplification [13,14], while the heterotrimeric G protein is the driving force for GPCRs to coordinate GTP phosphorylation of TCA substrates by the GTP–GDP exchange enzyme. Only when the mitochondria are fully utilized for oxidative phosphorylation can the true TCA cycle be realized. The anabolism required for cell reproduction will inevitably disrupt this cycle and produce lactate through aerobic glycolysis, which is then transported out of the cell (or tissue) and into the circulation [15].

Cancer cells need enough glucose to provide the carbon source for their growth and amino acids and nucleotides to provide the raw materials for protein and nucleic acid synthesis. At present, there are few reports on how cancer cells (or tissues) sense these nutrients and whether there is any difference in sensing capabilities between cancer cells and healthy cells.

It is well known that taste plays a key role in controlling the nutritional needs of animals. When we taste carbohydrates, we are rewarded with a sweet taste; when we taste nitrogen, including sodium glutamate or inosine when disodium acid is used, we are rewarded with umami. The known sweet taste receptors are T1R2-T1R3, and the umami taste receptors are T1R1-T1R3. Earlier studies showed that these taste receptors are found in taste bud tissues, but more recent studies have shown that these receptors are also present in other cells, tissues, and organs, particularly the stomach and intestines. These taste receptors are all G-protein-coupled receptors (GPCRs) [16]. Carbon from carbohydrates or nitrogen from amino acids and nucleotides transmits food signals to intracellular G proteins through appropriate receptor recognition and interrelated allosteric effects—after which, the G protein drives this signal through its GTP high-energy bond for activation, amplification, and delivery, thereby promoting cellular uptake, transport, and utilization.

Many research strategies have been established to study GPCRs, which are the most important targets for drug discovery [17]. For example, methods based on the live cell assay platform include well-plate HTS (high-throughput screening assays), cell microarrays, and cell microfluidics. Non-labeled methods (label-free systems) include surface plasmon resonance (SPR), resonant waveguide grating (RWG), impedance, and surface acoustic wave sensors (SAW) [18,19]. However, none of these reported methods could implement the measurements and comparisons of nutrient sensing kinetics between cancer cells and healthy cells. In this study, an electrochemical biological (tissue) sensor was developed by immobilizing colorectal cancer tissue and adjacent tissue (healthy tissue on the other side of the cancer tissue). This approach largely simulated the process by which tissues (or cells) sense and identify carbon and nitrogen nutrients. The biological signals were transmitted and recorded as electrochemical signals to facilitate the measurement and study of the sensing kinetics, the working principle of the prepared sensor, as shown in Figure 1. During the assay, carbon and nitrogen nutrients and GPCRs interact and ion channels open, causing Ca^2+^ flow and resulting in a change in membrane potential. The electrical signal output from the change in potential is detected by an electrochemical workstation and then analyzed for kinetic parameters to study the sensing ability of cancer and normal cells to carbon and nitrogen nutrients.

## 2. Results and Discussion

### 2.1. Potential Optimization of the Current-Time Measurement Method

The prepared sensor was measured by the current–time method at different potentials (the test substrate was ultrapure water). The steady-state current differences before and after the addition of 10^−5^ mol/L glucose were used to measure the influence of different potentials on the electrochemical responses of the sensor. The results showed that the value of the current change was the largest under the condition of −0.38 V, as shown in Figure 2A. Therefore, −0.38 V was chosen as the constant potential to investigate the response characteristics of the biosensor to glucose, sodium glutamate, disodium inosinate, and sodium lactate.

The redox peak potential difference is less than 80 mV, and the peak current ratio is close to 1, as shown in Figure 2B. As shown in Figure 2C,D, under different scanning rate conditions (25, 50, 75, 100, 125, 150, and 200 mV/s), the reduction peak and oxidation peak current of the electrode after activation have a good linear relationship with the square root of the scanning rate, indicating that the glassy carbon electrode after the treatment, the redox peak current, is controlled only by the diffusion conditions, and the pretreatment effect of the glassy carbon electrode is up to the standard, and subsequent research can be carried out.

As shown in Figure 2E, under the conditions of a scanning rate of 50 mV/s and scanning range of −0.1~0.6 V, the peak current value after the assembly of the nuclear microporous membrane is lower than that of the bare electrode. This is because the nuclear microporous membrane impedes the transfer of electrons to the electrode surface; the peak current value is further reduced after the assembly of the colon cancer tissue or adjacent tissue, because the colon cancer tissue or adjacent tissue impedes the electron transfer. This shows that the electrode assembly is successful.

### 2.2. Studies on the Sensing Kinetics of Carbon and Nitrogen Nutrients

#### 2.2.1. The Curve of Carbon and Nitrogen Nutrient Detection by the Cancer Adjacent Tissue-Based Biosensor

The immobilized electrochemical sensor of the cancer-adjacent tissues was placed in solutions of different concentrations of glucose, sodium glutamate, disodium inosinate, and sodium lactate ranging from a low concentration (10^−15^ mol/L) to a high concentration (10^−5^ mol/L) for time–current scanning. The scanning potential was −0.38 V. After complete incubation of the receptor and ligand, the 90th second current value was selected as the steady-state current. The rate of current change, ΔI, before and after receptor–ligand binding was plotted against the ligand concentration. For ease of plotting, the logarithmic concentration values of glucose, sodium glutamate, disodium inosinate, and sodium lactate were plotted on the abscissa, and the change rates of the response currents were plotted on the ordinate, as shown in Figure 3.

The data in Figure 3 show that the current change rates of glucose, sodium glutamate, disodium inosinate, and sodium lactate were stable at specific concentrations, and the kinetic curves were similar to enzymatic reactions [20]. In fact, when the ligand concentration was low, the signal output was proportional to the ligand concentration. When the ligand concentration reached a certain value, the signal output reached the maximum value. The signal output did not increase as the ligand concentration increased. Therefore, the kinetic analysis of the signal output generated by the interaction of carbon and nitrogen nutrients and their receptors could be carried out by referring to the kinetic characteristics and parameters of the enzyme reactions.

#### 2.2.2. Kinetic Curves of the Signal Output from the Interactions of Four Carbon and Nitrogen Nutrients with Receptors in Adjacent Tissues

As shown in Figure 3, the results showed that glucose, sodium glutamate, disodium inosinate, and sodium lactate were in the concentration ranges of 10^−15^ mol/L to 10^−12^ mol/L, 10^−14^ mol/L to 10^−12^ mol/L, 10^−15^ mol/L to 10^−13^ mol/L, and 10^−14^ mol/L to 10^−12^ mol/L, respectively. It was possible that there was a good increasing relationship with the rate of change of the response current. Therefore, the four carbon and nitrogen nutrients were further subdivided within their respective concentration ranges to evaluate the concentration of carbon nutrients (glucose and sodium lactate) and nitrogen nutrients (sodium glutamate and disodium inosinate) as the abscissa, and the change rates of the output currents were plotted on the ordinate, according to the enzyme–substrate catalysis kinetic formula [21]:[L] + [R] → [LR] → P (output) (1)
where [L] stands for ligand, [R] for receptor, and P for the electrochemical signal. The above formula indicates that the ligand and receptor interact to generate electrochemical signals and amplify the output. The curve obtained by hyperbolic fitting is shown in Figure 4.

When the receptor was saturated [22]:(2)$$\mathrm{Kd}=\frac{k1}{k2}=\frac{\left[R\right]\left[L\right]}{\left[\mathrm{RL}\right]}$$

If [RT] is set to the initial concentration of the receptor, [R] = [RT] − [RL]; if [LT] is the total ligand concentration, [L] = [LT] − [RL]. Substituting [L] = [LT] − [RL]/[R] = [RT] − [RL] into Equation (2) to obtain a new equation:(3)$${\left[\mathrm{RL}\right]}^{2}-\left[\mathrm{RL}\right]\left\{\left[\mathrm{RT}\right]+\left[\mathrm{LT}\right]+\mathrm{Kd}\right\}+\left[\mathrm{RT}\right]\left[\mathrm{LT}\right]=0$$

Equation (3) is a hyperbolic quadratic equation with an unknown quantity, and the variable is [RL]. When [RT]/Kd is a fixed value, [RL] will change with the change in [LT], rising rapidly at the beginning and then gradually reaching the level that is a saturation curve of receptor–ligand interaction. The equation showed that receptor–ligand binding had ligand saturation, similar to the enzyme–substrate interaction, and substrate saturation was the marker of enzyme catalysis.

When [R] = [RT] − [RL] was substituted into Equation (2), the resulting equation was as follows:(4)$$\mathrm{Kd}=\frac{\left\{\left[\mathrm{RT}\right]-\left[\mathrm{RL}\right]\right\}\left[L\right]}{\mathrm{RL}}$$

By rearranging Equation (4), the following double reciprocal equation is obtained:(5)$$\frac{1}{\left[\mathrm{RL}\right]}=\frac{1}{\left[\mathrm{RT}\right]}+\frac{\mathrm{Kd}}{\left[\mathrm{RT}\right]}\frac{1}{\left[L\right]}$$

The data from Figure 4 show that the interaction curves of the carbon nutrients (glucose and sodium lactate) and nitrogen nutrients (sodium glutamate and disodium inosinate) with their receptors conformed to a hyperbolic curve and had a ligand–receptor saturation effect similar to the enzyme–substrate action kinetics. According to the equation in Table 1, the reciprocal values of the concentrations of glucose, sodium glutamate, disodium inosinate, and sodium lactate were used as the abscissa, and the reciprocal values of the current change rates were plotted as the ordinate (i.e., the double reciprocal plot method) to obtain Figure 5. The double reciprocal fitting linear equation could then be obtained from Figure 5 to generate the data in Table 2.

#### 2.2.3. Kinetic Curves of the Signal Output from the Interaction of the Four Nutrients and the Receptor on Cancer Tissue

The method of measurement was similar to that used in the adjacent tissues. Firstly, the action curves of the detection range were plotted, from which the kinetic curves of the signal output through the interactions of the four nutrients and the receptors on the cancer tissues could be obtained. The results in Figure 6 show that the cancer tissue could be considered approximately as a baseline for sodium lactate, suggesting that the cancer tissue was not sensitive to sodium lactate, whereas the other three substances showed obvious increases and stable ranges. The same hyperbolic fitting and linear regression (Figure 6E–J) were performed to obtain the hyperbolic equation and linear regression equation (as shown in Table 3 and Table 4).

### 2.3. Activation Constant of the Receptor and the Ligand

The hyperbolic fit curve obtained from Figure 4 was very similar to the Michaelis–Menten equation for enzymatic reactions. Therefore, the reciprocals of the molar concentrations of the four substances and the reciprocals of the current rate of change could be fitted to the double reciprocal linear fit according to Figure 5 to obtain the linear regression equation, which calculated an activation constant Ka similar to the Michaelis constant (Table 5). The essence of this constant is the ligand concentration at which the interrelated allosteric effect generated by ligand and receptor recognition triggers activation and amplification of the intracellular signal to half the maximum signal output. In other words, the smaller the activation constant, the greater the sensitivity of the ligand to the receptor activation. From this, the response concentrations of neighboring tissues to different types of ligands could be deduced.

The results in Table 5 show that the activation constants differed between the adjacent tissues and the cancerous tissues. For three substances, including glucose, sodium glutamate, and disodium inosinate, the differences in Ka values between the cancer tissues and adjacent tissues were 14 times, 115 times, and 3 times, respectively. Sodium lactate, however, showed a different pattern; the cancer tissues had almost no response, whereas the adjacent tissues showed a strong sensing ability, with a sensitivity only lower than that of glucose. We were particularly interested in the way in which these dramatic differences in the sensing capabilities of the cancer and adjacent tissues were expressed. We proposed two possible explanations for this: differences in the number of receptors on the cell surface and changes in the intracellular cascade amplification system. We estimated the intracellular amplification of the signaling cascade and the minimum number of receptors required to activate the maximum output signal on the cell surface to identify the causes of these differences.

### 2.4. Estimation of Cell Cascade Magnification

The common feature of GPCRs is the seven-transmembrane structure. If the process of recognition between the extracellular domain and the ligand occurs through a “signature” method—namely, through the interconnected allosteric interactions between the seven transmembrane domains—the signature is in the G protein activation mode [23] and the simultaneous activation of genomic and non-genomic pathways. The kinetic laws and parameters determined in this study included the control of plasma membrane ion channels by non-genomic signaling pathways by G proteins. Studies have shown that the ligands used in this study mainly sense and amplify signals through T1R1 and T1R2. The process of receptors and ligand recognition activated the heterotrimeric G proteins through interrelated allosteric effects, and activation of the βγ subunit additionally activated PLCβ2, causing Ca^2+^ to flow in from the transient receptor potential ion channel protein 5 (TRPM5), resulting in the depolarization of the cell. These changes in electrochemical signals can be detected and collected by the glassy carbon electrode. We were also able to quantitatively measure and estimate these electrical signal changes using bare electrodes as a control. Similarly, the logarithms of the concentrations of the four substance solutions were used as the abscissa, and the current change rates were plotted on the ordinate to perform a linear fit. The interaction equations of the four ligands of glucose, sodium glutamate, disodium inosinate, and sodium lactate with the bare electrode were as described below.

The bare electrode concentrations could be calculated by using the current change rates corresponding to the activation constant Ka in Table 5 in the bare electrode equation in Table 6. The following equations could then be used to calculate the cascade amplification folds [23]:M = 10[lg(Ca) − lg(Ka)] (6)

It is obvious that there was a significant difference in the amplification factor of the intracellular signals between the cancerous tissues and the peripheral cancerous tissues. The amplification factor of the intracellular signals in the cancer tissues was much higher than that of the peripheral cancer tissues. The differences in the signal amplification of glucose, sodium glutamate, and disodium inosinate were about 249 times, 270 times, and 16 times, respectively. The experimental results confirmed that, in comparison with the cancer-adjacent tissues, the colon cancer cells required nutrients for their reproduction, including glucose, sodium glutamate, and disodium inosinate. The increased sensing ability was primarily manifested by increased activation of the intracellular signal amplification system.

### 2.5. Estimation of the Minimum Number of Receptors Required to Activate Cancer-Adjacent Tissues or Cancerous Tissues to Achieve the Maximum Signal Output

Through the kinetic equation of ligand–receptor interactions, the minimum number of receptors (hereafter referred to as the minimum number of receptors) required to activate the adjacent or cancerous tissue to obtain the maximum signal output could be calculated. A reported method [24] defined the minimum number of receptors required to activate the adjacent tissue or cancerous tissue to obtain the maximum signal output as follows:(7)$$n=\frac{C_{M}}{C_{L}}=\frac{2Ka}{C_{L}}$$

In the equation, C_M_ is the concentration that gave the maximum response signal when the number of receptors on the cell was not one, and C_L_ is the concentration that gave the maximum response signal when there was only one receptor on the cell surface. After careful calculation, it was concluded that the minimum number of receptors for glucose, sodium glutamate, disodium inosinate, and sodium lactate on the colon cancer tissue cells were 1.49, 2.20, 9.21, and 0, respectively. The minimum number of receptors for glucose, sodium glutamate, disodium inosinate, and sodium lactate on the healthy cells adjacent to the cancer tissues were 2.02, 2.55, 2.85, and 1.83, respectively. The results showed that the number of receptors for the two substances of glucose and sodium glutamate did not differ much between the cancer tissues and the peripheral cancer tissues, whereas disodium inosinate was significantly different. Studies have reported that T1R1 can sense sodium glutamate and disodium inosinate, and another receptor, the metabotropic glutamate receptors (mGluR) [25], can only sense glutamate sodium but not inosine. Therefore, based on the above information, we speculated that the increased sensitivity of colon cancer tissue to disodium inosinate would mainly depend on the increased intracellular signal amplification of disodium inosinate through T1R1 activation rather than the increased signal amplification system of inosine. In other words, the signal amplification systems for glucose and sodium glutamate showed a higher degree of improvement than disodium inosinate. By estimating the amplification of the cell signal cascade, the minimum number of receptors on the cell surface, it is clear that the sensing of glucose and sodium glutamate by the cancer tissues was primarily achieved by altering the intracellular signal amplification.

### 2.6. Stability of Electrochemical Sensor

The assembled electrochemical sensor was continuously tested 10 times in 10^−5^ mol/L glucose solution, and the relative standard deviation of the current rate of change was 4.68%, indicating that the stability of the sensor was good. The sensor was suspended above the PBS buffer solution and stored at 4 °C. It was measured daily in the same concentration of glucose solution, and the response current value of this sensor was relatively stable for the first 8 days. The response current value on day 8 was 85.45% of that on day 1. This indicates that the receptor sensor can be stored stably for at least 8 days.

### 2.7. Discussion

Indeed, the cancerous and peripheral cancerous tissues differed in their ability to sense carbon and nitrogen nutrients, whereas the cell surface receptors of the healthy and cancerous cells remained unchanged, eliminating the differences in ligand–receptor interactions. Therefore, our study focused only on intracellular signaling. Both sweet taste sensor receptors T1R2/T1R3 and umami taste sensor receptors T1R1/T1R3 are G protein-coupled receptors. Carbohydrates bind to T1R2/T1R3 to activate α-tastein, thereby activating adenosine acid cyclase (adenylyl cyclase, AC) to produce 3′, 5′-cyclic adenylate (cAMP), leading to an increase in intracellular cAMP concentration. This led to the activation of protein kinase A (PKA) and K^+^ phosphorylation of the channel to close the ion channel, inhibiting K^+^ outflow, membrane depolarization, and neurotransmitter release. The umami taste stimulant bound to T1R1/T1R3 to activate α-tastein, resulting in the separation of the β and γ subunits of the G protein. The released Gβ3 and Gγ13 subunits activated phospholipase-β2 (phospholipase-β2, PLC-β2), and PLC-β2 hydrolyzed phosphatidylinositol-4,5-bisphosphate [PI(4,5) P2] into diacylglycerol (DAG) and inositol triphosphate (IP3), which binds to the third type of inositol triphosphate receptor 3 (IP3R3), resulting in the opening of the IP3-gated Ca^2+^ channel on the intracellular organelle membrane and the intracellular Ca^2+^ storage. In turn, the increase in the concentration of Ca^2+^ in the cytoplasm led to the transient receptor potential melastatin 5 (TRPM5) channel to open, resulting in the inflow of Na^+^ to enter the cell. This will ultimately lead to membrane depolarization and neurotransmitter release, thereby generating electrical signals.

G proteins use the energy released by the hydrolysis of GTP to GDP to control the switching of ion channels. While controlling the rapid transmission of electrochemical signals, they also control the replication, transcription, internal and external materials, and energy of the genome through complex signaling pathways, nuclear factor and transcription factor networks, and energy and signal exchange [26]. They also dominate the cell cycle, proliferation, and autophagy by controlling write; read; and erase (cell energy charge, reduction power, and modification of metabolic intermediates) on nucleosomes and apoptosis [27].

The main difference between the cancerous and cancer-adjacent tissues in this study was the intracellular signal amplification. Although a rapid non-genomic approach was used for measurements, the difference in the signal amplification system was determined by the complex epigenetic modification of the colorectal cancer cells and the peripheral cancer tissue cells, as the fast pathway of colorectal cancer cell progression remained the same. Compared to healthy cells, the biggest difference in cancer cells is that they proliferate indefinitely [28]. The premise is that they must continuously sense and absorb glucose to provide themselves with carbon skeletons and raw materials and thus continuously sense amino acids and nuclear glycolic acid to provide nitrogen nutrition for DNA replication, transcription, and protein synthesis. However, nucleotide, amino acid, and fatty acid synthesis should be involved in the TCA cycle, thereby interrupting the transfer of hydrogen protons to oxygen through the oxidized respiratory chain, interrupting oxidized phosphoric acid. The hydrogen protons can then only be transferred to pyruvate to produce lactate or removed from cells or tissues by the Warburg effect. Healthy cells undergo oxidative phosphorylation to obtain energy from the bloodstream [29]. Recent studies have consistently shown that all cells must undergo these regulatory transformations in the process of proliferation. In particular, the proliferation of immune cells also exhibits a Warburg effect [30], whereas the proliferation of healthy cells can be controlled. Specifically, only when cells need to proliferate do they continuously sense and absorb the nutrients necessary for their proliferation, and they will not continue to do so when they do not need to proliferate. Therefore, the manner by which healthy cells stop sensing and absorbing the nutrients necessary for their proliferation is an issue that must be clarified. Based on the results of a large number of recent studies, it is clear that the sensory control of essential nutrients is the key to this problem. The sensory control of cells is similar to the control of animal nutrition. Under starvation conditions, carbohydrates are rewarded with sweetness and amino acids, or nucleotides are rewarded with umami. In contrast, in satiated conditions, even if these nutrients and receptors are still present, they are no longer rewarded in the same way as when we are hungry. It is clear that the key to its control (especially negative inhibition) is not the receptor or the ligand but the control of the intracellular signal amplification process. The results of this study showed that the major differences between the colon cancer tissues and adjacent tissues were in the constitutive and heritable differences in signal transmission and amplification. In this regard, it could be inferred that continuous proliferation depends on continuous nutrient sensing and absorption, and continuous nutrient sensing and absorption depend on the continuous transcription and expression of the corresponding genes. Simply put, epigenetic modifications to the lysine side of nucleosome caused by repeated writing, reading, and erasing inhibit the energy charge (which is ATP) required to support cell proliferation; key metabolic intermediates (including lactate); coenzymes (including coenzymes I, II, and acetyl-CoA); heterotrimeric G protein; and the GTP substrate-level phosphorylation substrates (succinyl-CoA) required by the small G protein [27] are the fundamental cause of cancer initiation and development. In general, the difference in carbon and nitrogen nutrition between cancer cells and para-cancerous cells is an illusion, and the reason lies in the changes in intracellular metabolism and the result of metabolic reprogramming. GPCRs play a key role in this process, accompanied by the exchange of GDP-GTP and ATP-cAMP, the phosphorylation of various kinases, and the activation of multiple downstream pathways by small G proteins. At the same time, the metabolism also changes. A large number of intermediate metabolites in the TCA cycle accumulate, are excreted from the mitochondria, and participate in anabolic metabolism, providing the proteins, lipids, and ribose required for the proliferation of cancer cell proliferation. Finally, the TCA cycle is disrupted, and glutamine is produced. A variety of coenzymes such as acetyl, succinyl, and malonyl also accumulate during this process to modify histones and feed back into the G protein signal amplification system. The intracellular signal amplification coefficient is increased, allowing lower concentrations of carbon and nitrogen nutrients to be taken up (Figure 7).

Among the four carbon and nitrogen nutrients, sodium lactate is unique. Lactate can be transported by the monocarboxylate transporter (MCT) in tissues, cells, and organs and regulates body synthesis and catabolism in the body, where it plays a key role in the process. Lactate can also be used as a carbon source by the human body [31,32]. In addition, lactic acid has a specific sensor receptor, the HCAR1 receptor, through which lactic acid can directly inhibit the action of the NLRP3 inflammasome, which is activated by the action of TLR-4 and caspase1. As a result, the effect of NF-κB is reduced, and the conversion of PRO-IL-1b to active IL-1b is reduced [33,34]. This has a significant beneficial effect on disease severity in these two models. Lactic acid at physiological concentrations is a good anti-inflammatory agent, and this anti-inflammatory effect can effectively inhibit cell reproduction. However, the cancer tissues rejected the lactate. One of the most striking features of cancer cells is the Warburg effect, or aerobic glycolysis. This process produces a large amount of lactic acid, which is released into the circulation in the form of salt and is sensed, absorbed and utilized by healthy cells. For example, the TCA cycle is completely oxidized. For healthy cells, lactate is both a carbon nutrient and an energy substance, whereas, for cancerous tissues, which produce large amounts of lactic acid, lactate is no longer a nutrient. The loss of lactate-sensing ability in cancer cells is clearly an inevitable result, which was confirmed in this study.

## 3. Materials and Methods

### 3.1. Materials and Reagents

Soluble starch was purchased from Yingda Rare Chemical Reagents (Tianjin, China). Sodium alginate was purchased from Guangfu Fine Chemical Research Institute (Tianjin, China). CaCl_2_ was purchased from Sigma (USA). Glutaraldehyde was purchased from Bodi Chemicals (Tianjin, China). Microporous membrane was a product of Whatman (UK). NaCl was purchased from Amresco (USA); 1.00, 0.30, and 0.05 μm diameter α-Al_2_O_3_ and suede were purchased from Zhenhua Instrument (Shanghai, China). Absolute ethanol, HNO_3_, H_2_SO_4_, KNO_3_, and K_3_Fe(CN)_6_ were purchased from Guangfu Reagents (Tianjin, China). DMEM, fetal bovine serum, antibiotics, urethane, and trypan blue were purchased from Sigma. The CT26 mouse colon cancer cell line was purchased from Zhongqiao Xinzhou (Shanghai, China). Male BALB/c mice were obtained from Aoyide Experimental Supplies (Tianjin, China). All reagents were of analytical grade. Ultrapure water was used throughout the entire experiment.

### 3.2. Instruments and Equipment

The following instruments were used in this study: An analytical balance(Precision Scientific Instrument (Shanghai, China)), Milli-Q Ultrapure Water System (Yarong Biochemical Equipment (Shanghai, China)), KQ 3200B Ultrasonic cleaner (Ultrasonic Instrument (Kunshan, Jiangsu, China)), water bath (Yiheng Technology (Shanghai, China)), CHI600E Electric ChemStation (Chenhua Instrument (Shanghai, China)), and three-electrode system (glassy carbon electrode Φ = 3 mm, Ag/AgCl electrode, platinum wire electrode; Chenhua Instrument).

### 3.3. Methods

#### 3.3.1. Establishment of the Mouse Colon Cancer Model

The mouse colon cancer model was adopted from the non-surgical anorectal injection method of Donigan et al. [35]. Firstly, CT26 mouse colon cancer cells were cultured in DMEM, fetal bovine serum, and antibiotics at the ratio of 8:1:0.5 in a cell culture incubator. After three passages of subculture, cell viability was assessed using the trypan blue staining method. The final cell suspension with 90% cell viability was used for animal injection.

Male Balb/c mice were anesthetized by intraperitoneal injection of 10% urethane solution (0.75 g/kg). Subsequently, the mice were then gently dilated at the anal opening using blunt-tipped forceps. CT26 cells (2.5 × 10^4^) suspended in DMEM containing 10% FBS were injected submucosally into the distal posterior rectum using a syringe with a 29 G needle, and the mice were euthanized 17 days after injection. Mice with a successfully established colon model were selected for immediate dissection of their colon cancer tissues and adjacent tissues. The obtained tissues were thoroughly rinsed to remove the contents, cut into a 0.25 cm^2^ size plane square, and stored in saline for later use. Tissues were obtained immediately from euthanized mice to ensure tissue freshness and bioactivity.

#### 3.3.2. Electrode Pretreatment and Effect Characterization

The glassy carbon electrode was successively polished on the suede leather with aluminum powder (α-Al_2_O_3_) with particle sizes of 1.0, 0.3, and 0.05 μm diameter. After each polishing, the electrode was cleaned in an ultrasonic bath for 30 s. The electrode cleaning procedure was repeated at least three times. The treated electrode surface was further washed with 50% HNO_3_, absolute ethanol, and ultrapure water in a sequential order. After cleaning, the electrode was immersed in a 1 mol/L H_2_SO_4_ solution with a scanning range of −1.0–1.0 V. The scanning rate of the cyclic voltammetry was set to 100 mV/s to activate the electrode.

After treatment, the electrode was immersed in a 1 mmol/L K_3_Fe(CN)_6_ solution containing 0.20 mol/L KNO_3_, and cyclic voltammetry was set at a scanning rate of 50 mV/s and a scanning range of −0.1–0.6 V to characterize the pretreatment effect of the glassy carbon electrode. After the pretreatment, the peak potential difference of the cyclic voltammetry curve of the glassy carbon electrode was less than 80 mV, and the electrode was ready for use. After detection, the electrode was cleaned with ultrapure water and then dried in a nitrogen environment for later use.

#### 3.3.3. Preparation of Electrochemical Biosensors

Soluble starch was dissolved in an aqueous solution containing 1% glutaraldehyde and heated in a water bath at 80 °C followed with stirring for 30 min to produce a starch solution with a specific mass concentration. The starch solution was allowed to settle overnight at room temperature overnight to fully cross-link the starch with glutaraldehyde to obtain an aldehyde-based starch gum solution. The aldehyde-based starch gum solution was mixed with a specific mass concentration of sodium alginate solution in a volume ratio of 1:1 [36,37,38]. Ten microliters of the above mixed solution was spread evenly on two microporous polycarbonate membranes with a diameter of 25 mm and a pore size of 0.22 μm. The 0.25 cm^2^ colon tissue (a peripheral cancerous tissue) immersed in saline was prepared and placed in the center of a microporous membrane and then covered with another measuring membrane to form a sandwich structure.

The prepared colon tissue and cancer tissue measuring membrane were immersed in a 5% CaCl_2_ solution by mass for 10 s before removal. This procedure ensured that the sodium alginate and CaCl_2_ underwent an ion-exchange reaction to form a stable chelate, and the sodium alginate solution was gelled into a good fixative [39,40]. The colon cancer tissue (the peripheral cancer tissue) was then rinsed with saline to remove the Cl^−^ and Ca^2+^, etc. remaining on the membrane. Finally, the tissue component of the prepared sandwich structure membrane was aligned with the glassy carbon electrode so that the colon cancer tissue (the peripheral cancer tissue) overlapped with the characterizing electrode core. The whole assembly was then fixed with a leather sheath to prepare a biosensor.

#### 3.3.4. Determination of the Electrochemical Biosensor on Mouse Colon Tissue

A three-electrode system was used, with the glassy carbon electrode fixed to the measuring membrane of mouse colon cancer tissue (the peripheral cancer tissue) as the working electrode. The Ag/AgCl electrode was used as the reference electrode, and a platinum wire electrode was used as the counter electrode. Ultrapure water was used as the blank. The response currents of sodium glutamate, disodium inosinate, glucose, and sodium lactate solutions at concentrations of 10^−15^–10^−4^ mol/L were determined by the current-time determination method under optimized voltage (−0.38 V). Each concentration was measured three times in parallel and the rate of change of the response currents were used as an indicator. The response current change rate, Equation (8), was
(8)$$\Delta I=\frac{I_{1}-I_{2}}{I_{1}}\times 100$$
where I_1_ and I_2_ represent the steady-state current values before and after the ligand compound was measured at the same time point, respectively.

## 4. Conclusions

In this study, the sensing kinetics of four different carbon and nitrogen nutrients, namely glucose, sodium glutamate, disodium inosinate, and sodium lactate, were investigated by establishing a biosensor for mouse colon cancer tissue and adjacent healthy tissue. The degree of amplification of the cell signal cascade and the minimum number of receptors required to activate cancerous or adjacent tissues for maximum signal output were calculated. The results showed that the sensing of the colon cancer tissue was mainly due to the constitutive adaptation of the cell signal amplification system of the carbon and nitrogen nutrient sensing receptors, which greatly improved the sensitivity to glucose, sodium glutamate, and disodium inosinate. However, the cancer cells lost the ability to sense lactate, so they could continue to take up and use these nutrients to protect themselves from unlimited proliferation. Meanwhile, the shutdown of the lactate sensing mechanism ensured its synthesis and metabolism, disrupting the energy supply of oxidative phosphorylation and eliminating lactate.

The tissue biosensor and corresponding method established in this study were easy to operate, highly sensitive, and reproducible. They could realize the receptor–ligand recognition, interrelated allosteric effect and the determination of the kinetic law of induced cell signaling cascade amplification, which may be clinically useful by providing new methods for diagnosis and screening of anticancer drugs. The study also showed for the first time that colon cancer tissue was insensitive to lactate, while the healthy tissue adjacent to the colon cancer was sensitive to lactate, suggesting that lactate may have potential applications in the clinical nutritional adjuvant treatment of colon cancer. Consumption of lactate by colorectal cancer patients may provide a carbon source for healthy gastrointestinal cells, while avoiding the sensing, absorption, and utilization of lactate by colorectal cancer tissues.

## Figures and Tables

**Figure 1 molecules-28-01453-f001:**
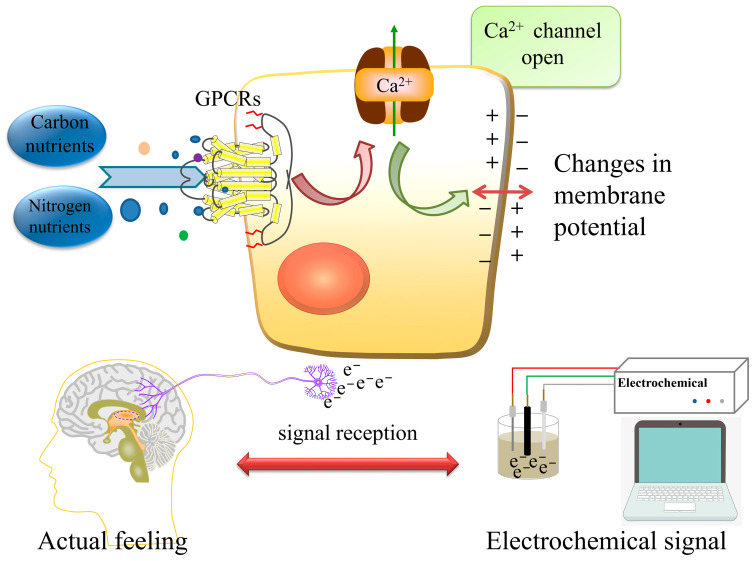
Principle of the electrochemical biological (tissue) biosensor. After carbon and nitrogen nutrients are incubated with taste receptors, the Ca^2+^ channel is opened to create a potential difference between the intracellular and extracellular compartments, which generates electrical signals from impulses. Collecting signals from the electrochemical workstation can truly simulate signal transmission in the human body and signal amplification in cells.

**Figure 2 molecules-28-01453-f002:**
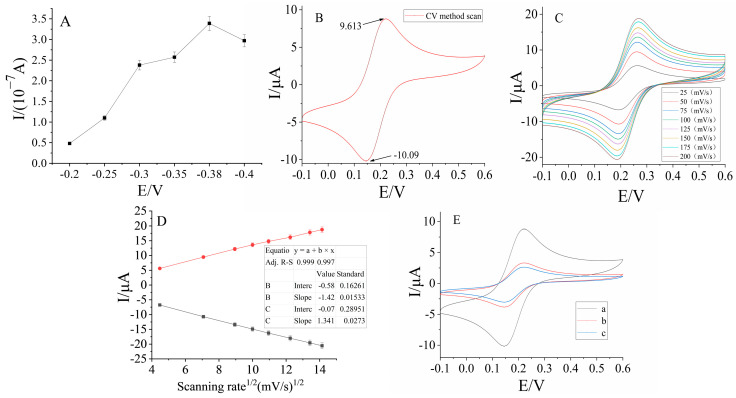
Electrochemical characterization of pretreated glassy carbon electrodes and potential optimization: (**A**) Effect of response potential on the biosensor. (**B**) The cyclic voltammogram of the bare electrode in the voltage range of −0.1~0.6 V with a scan rate of 50 mV/s. (**C**) Characterization diagram of cyclic voltammetry at different rates (25, 50, 75, 100, 125, 150, 175, and 200 mV/s). (**D**) The relationship between the square root of the scan rate and the redox peak current of the cyclic voltammetry curve at different scan rates. (**E**) Cyclic Voltammetry Characterization of Different Modified Sections of Electrodes. (a) Bare electrode. (b) Bare electrode + Nuclear microporous membrane. (c) Bare electrode + Nuclear microporous membrane + Colon cancer tissue/adjacent normal tissues.

**Figure 3 molecules-28-01453-f003:**
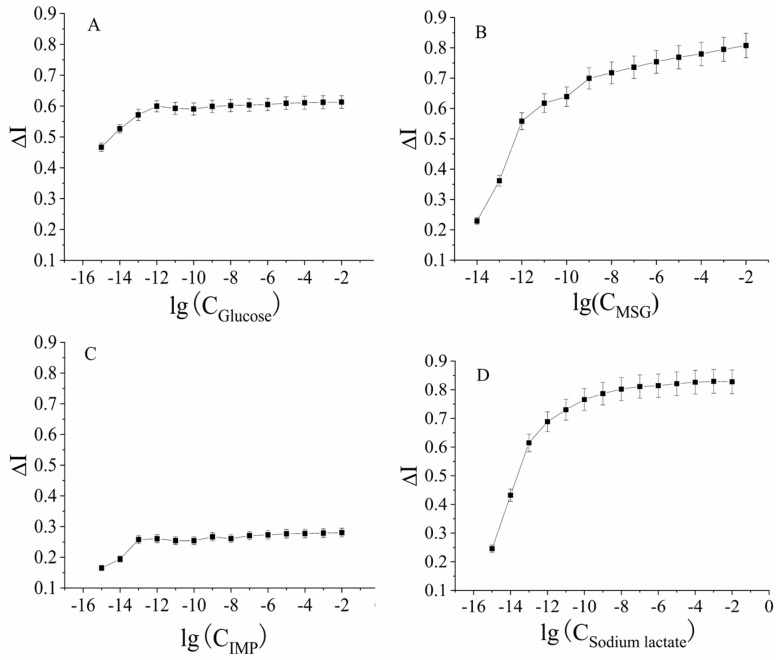
The response current curve of the mouse adjacent normal tissues sensor in different carbon and nitrogen nutrient concentration ranges: (**A**) Glucose solution within the concentration range of 10^−15^–10^−6^ mol/L. (**B**) Sodium glutamate solution within the concentration range of 10^−14^–10^−5^ mol/L. (**C**) Disodium inosinate solution in the concentration range of 10^−15^–10^−6^ mol/L. (**D**) Sodium lactate solution in the concentration range of 10^−15^–10^−6^ mol/L.

**Figure 4 molecules-28-01453-f004:**
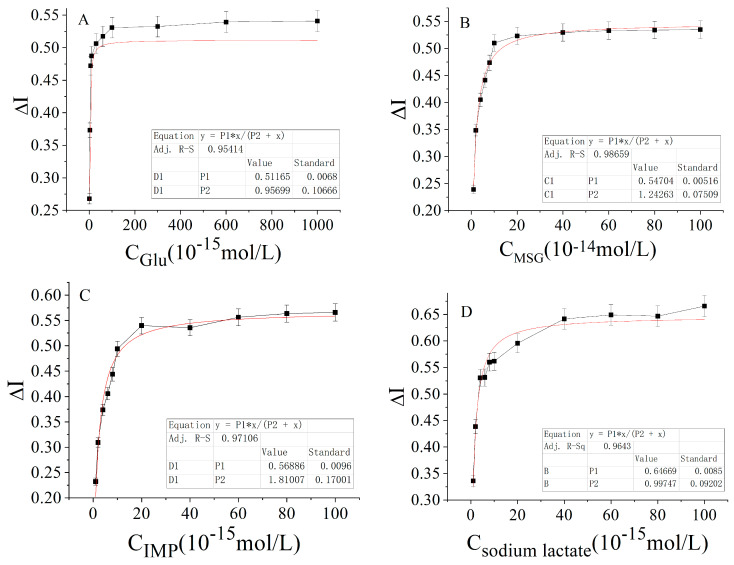
Rate of the current change and hyperbola fitting of mouse adjacent normal tissue sensor to different carbon and nitrogen nutrients: (**A**) Glucose solution within the concentration range of 10^−15^–10^−12^ mol/L. (**B**) Sodium glutamate solution within the concentration range of 10^−14^–10^−12^ mol/L. (**C**) Disodium inosinate solution in the concentration range of 10^−15^–10^−13^ mol/L. (**D**) Sodium lactate solution in the concentration range of 10^−15^–10^−13^ mol/L.

**Figure 5 molecules-28-01453-f005:**
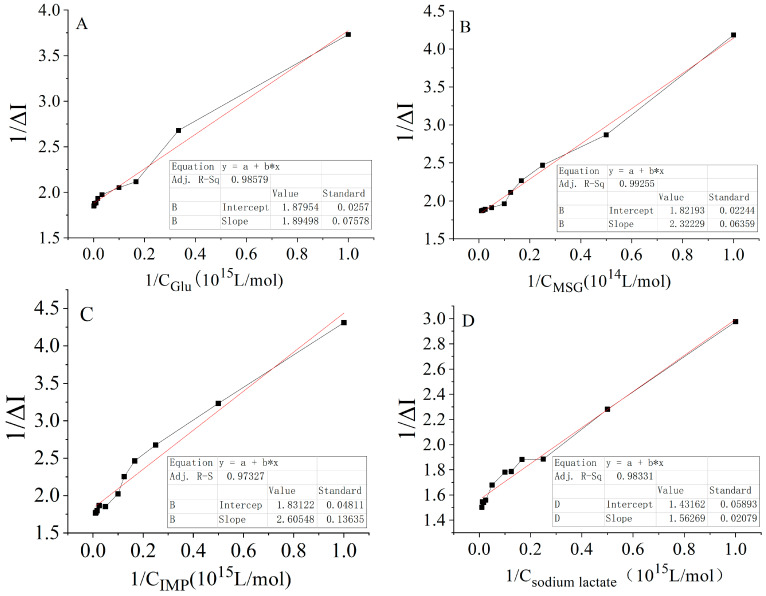
Double reciprocal curve of the rate of change of the sensor concentration and current in mouse adjacent normal tissue: (**A**) Glucose solution. (**B**) Sodium glutamate solution. (**C**) Disodium inosinate solution. (**D**) Sodium lactate solution.

**Figure 6 molecules-28-01453-f006:**
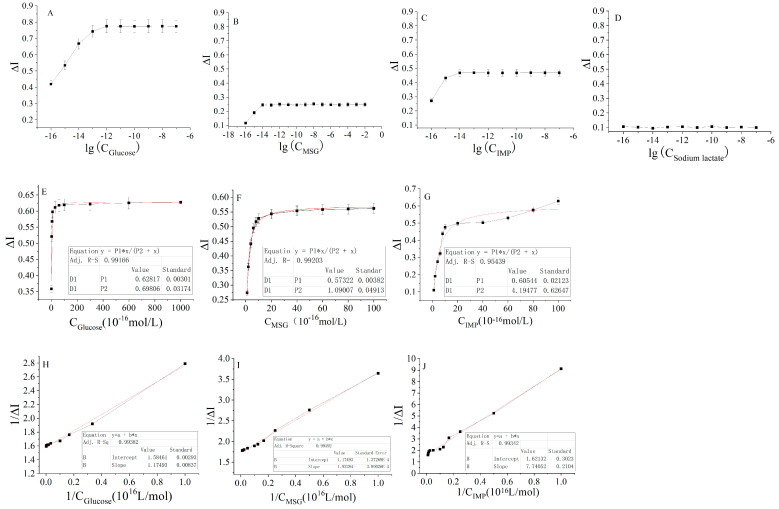
The response current curve of the mouse cancer tissue sensor in different nutrient concentration ranges of carbon and nitrogen: (**A**) Glucose solution in the concentration range of 10^−16^–10^−7^ mol/L. (**B**) Sodium glutamate solution within the concentration range of 10^−16^–10^−7^ mol/L. (**C**) Disodium inosinate solution in the concentration range of 10^−16^–10^−7^ mol/L. (**D**) Sodium lactate solution has no response concentration. Fitting of the response curve of the mouse colon cancer tissue sensor to glucose (**E**), sodium glutamate (**F**), and disodium inosinate (**G**) in the concentration range. Double reciprocal curve of the concentration–current change rate of the mouse colon cancer sensor for glucose (**H**), sodium glutamate (**I**), and disodium inosinate (**J**).

**Figure 7 molecules-28-01453-f007:**
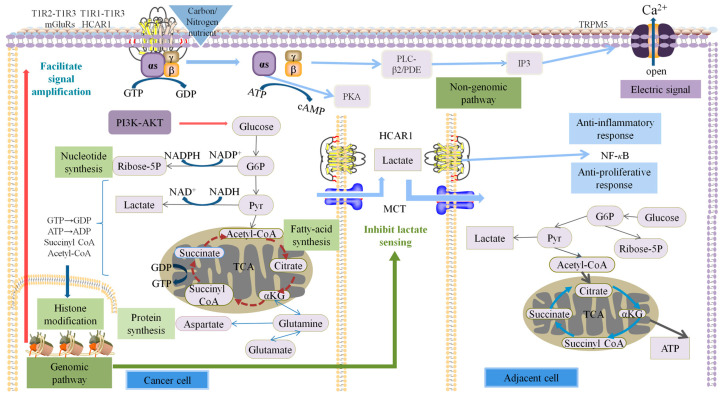
Differences in the sensing of carbon and nitrogen nutrients between cancerous tissues and adjacent normal tissues. When carbon and nitrogen nutrients bind to the corresponding receptors (G protein-coupled receptors), intracellular electrical signals are generated that are present in both cancerous and para-cancerous cells, namely: non-genomics. This is also the measuring principle of this experiment. G protein-coupled receptors then activate the downstream PI3K-AKT pathway, and at the same time, metabolic changes occur, and a variety of intermediate metabolites accumulate, providing raw materials for cancer cell proliferation. Succinyl-CoA, acetyl-CoA, etc., which accumulate in the metabolic process undergo histone modification, promotes the amplification of intracellular signals and improves the sensing of carbon and nitrogen nutrition. What is unique is that cancer cells inhibit the ability to sense lactate, while neighboring cells can still sense and use lactate.

**Table 1 molecules-28-01453-t001:** Hyperbolic equations for the interactions of four carbon and nitrogen nutrients with receptors in adjacent normal tissues.

Carbon and Nitrogen Nutrients	Hyperbolic Equation	Correlation Coefficient (R^2^)
Glucose	ΔI = 0.51165 (±0.0068) × 10^−15^ C/(0.95699 (±0.10666) + C × 10^−15^)	0.9541
MSG	ΔI = 0.54704 (±0.00516) × 10^−14^ C/(1.24263 (±0.07509) + C × 10^−14^)	0.9866
IMP	ΔI = 0.56886 (±0.0096) × 10^−15^ C/(1.81007 (±0.17001) + C × 10^−15^)	0.9711
Sodium lactate	ΔI = 0.64669 (±0.0085) × 10^−14^ C/(0.99747 (±0.09202) + C × 10^−14^)	0.9643

**Table 2 molecules-28-01453-t002:** Linear regression equations for the interactions of four carbon and nitrogen nutrients with receptors in adjacent normal tissues.

Carbon and Nitrogen Nutrients	Linear Regression Equation	Correlation Coefficient (R^2^)
Glucose	1/ΔI = 1.8950 (±0.07578) × 10^15^ 1/C + 1.8795 (±0.0257)	0.9858
MSG	1/ΔI = 2.3223 (±0.06459) × 10^14^ 1/C + 1.8219 (±0.02244)	0.9926
IMP	1/ΔI = 2.6055 (±0.13635) × 10^15^ 1/C + 1.8312 (±0.04811)	0.9732
Sodium lactate	1/ΔI = 1.5627 (±0.02079) × 10^14^ 1/C + 1.4316 (±0.05893)	0.9833

**Table 3 molecules-28-01453-t003:** Hyperbolic equations for the interactions between four carbon and nitrogen nutrients and cancer tissue receptors.

Carbon and Nitrogen Nutrients	Hyperbolic Equation	Correlation Coefficient (R^2^)
Glucose	ΔI = 0.62817 (±0.00301) × 10^−16^ C/(0.69806 (±0.03174) + C × 10^−16^)	0.9917
MSG	ΔI = 0.57322 (±0.00382) × 10^−16^ C/(1.09007 (±0.04913) + C × 10^−16^)	0.9920
IMP	ΔI = 0.60544 (±0.02123) × 10^−16^ C/(4.19477 (±0.62647) + C × 10^−16^)	0.9544
Sodium lactate	___	___

**Table 4 molecules-28-01453-t004:** Linear regression equation for the interactions between four carbon and nitrogen nutrients and cancer tissue receptors.

Carbon and Nitrogen Nutrients	Linear Regression Equation	Correlation Coefficient (R^2^)
Glucose	1/ΔI = 1.1794 (±0.00293) × 10^16^ 1/C + 1.5846 (±0.00837)	0.9938
MSG	1/ΔI = 1.9228 (±0.00039) × 10^16^ 1/C + 1.7429 (±0.00014)	0.9959
IMP	1/ΔI = 7.4705 (±0.2104) × 10^16^ 1/C + 1.6213 (±0.3023)	0.9934
Sodium lactate	___	___

**Table 5 molecules-28-01453-t005:** The activation constants, cascade magnification, and minimum number of receptors of the four carbon and nitrogen nutrients and adjacent and cancerous tissues.

Carbon and Nitrogen Nutrients	Activation Constant (Ka)		Cascade Magnification		Minimum Number of Receptors	
	Colon Cancer Tissue	Adjacent Tissues	Colon Cancer Tissue	Adjacent Tissues	Colon Cancer Tissue	Adjacent Tissues
Glucose	7.438 × 10^‒17^	1.008 × 10^‒15^	1.544 × 10^4^	61.975	1.49	2.02
MSG	1.103 × 10^‒16^	1.275 × 10^‒14^	2.322 × 10^4^	85.883	2.20	2.55
IMP	4.608 × 10^‒16^	1.423 × 10^‒15^	1.943 × 10^5^	1.242 × 10^3^	9.21	2.85
Sodium lactate		9.162 × 10^‒16^		2.128 × 10^6^		1.83

Note: We also calculated the Ka value by nonlinear fitting of the Hill function in origin9.0. The results are basically consistent with the Ka value obtained by double reciprocal linear fitting in this study. The results of the Ka value calculated by nonlinear fitting are as follows: adjacent normal tissues: The allosteric constant Ka’ values corresponding to interactions with glucose, sodium glutamate, disodium inosinate, and sodium lactate are 1.042 × 10^−15^, 1.252 × 10^−14^, 1.857 × 10^−15^, and 9.261 × 10^−16^ mol/L; colon cancer tissues: The allosteric constant Ka’ values corresponding to interactions with glucose, sodium glutamate, and disodium inosinate are 7.696 × 10^−17^, 1.1167 × 10^−16^, and 4.032 × 10^−16^ mol/L. The correlation coefficients (R^2^) of nonlinear fitting are all greater than 0.95.

**Table 6 molecules-28-01453-t006:** The interaction equations of four carbon and nitrogen nutrients and bare electrodes.

Carbon and Nitrogen Nutrients	Bare Electrode Action Equation	Correlation Coefficient (R^2^)
Glucose	∆I/% = 5.02508 C_1_ + 92.4035	0.9914
MSG	∆I/% = 4.76044 Ca + 92.27473	0.9872
IMP	∆I/% = 3.90549 C_3_ + 70.93407	0.9891
Sodium lactate	∆I/% = 5.4711 C_4_ + 100.4991	0.9689

## Data Availability

This article has not been submitted to other journals, and the cited materials are labeled references.

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
