# Peer review of "Comparative Study on the Sensing Kinetics of Carbon and Nitrogen Nutrients in Cancer Tissues and Normal Tissues Based Electrochemical Biosensors"

_molecules, 2023, doi:10.3390/molecules28031453_

Round 1

Reviewer 1 Report

The authors have a done a good job characterizing and discussing the potential differences in sensing of sodium and carbon nutrients in cancer vs normal cells. Minor changes are suggested: 

1. Fig 2A: What is this graph of? They mention they test all the glucose, sodium glutamate, disodium inosinate, and sodium lactate, but do not mention which one of these is plotted in the figure.

2. There are no error bars in Figs 2, 3 and 4. 

3. Section 3.2 (Line 232-233): The current-time response for all the nutrients should be provided in the manuscript.

Author Response

The authors have a done a good job characterizing and discussing the potential differences in sensing of sodium and carbon nutrients in cancer vs normal cells. Minor changes are suggested:

A: Thank you very much for your evaluation and recognition about our work. We have revised the following questions carefully according to your suggestion.

Q1: Fig 2A: What is this graph of? They mention they test all the glucose, sodium glutamate, disodium inosinate, and sodium lactate, but do not mention which one of these is plotted in the figure.

A: Thanks to the reviewer for the reminder. Figure 2A shows the rate of change of the sensor current at different voltages, measuring the difference in steady-state current before and after the addition of 10-5 mol/L glucose, which was used to measure the effect of different potentials on the effect of the electrochemical response of the sensor.

Q2: There are no error bars in Figs 2, 3 and 4.

A: Thanks reviewer. Error bars have been added to Figs2,3 and 4.

Q3: Section 3.2 (Line 232-233): The current-time response for all the nutrients should be provided in the manuscript.

A: Thanks reviewer. The response current change rate curves of mouse adjacent normal tissues and mouse cancer tissues to four carbon and nitrogen nutrition in different concentration ranges have been shown in Fig.3 and Fig.6, respectively.

Reviewer 2 Report

The study is very interesting, relevant, and of general interest to the readers of this journal.

The introduction part covers both old and new references in a succinct way and has perfect integration of the main aspects of the theme.

The cited core references are recent and appropriate to the discussion.

This article is well written, with a good organization of the contents and a clear and pertinent methodology, particularly the biosensor construction approach. Nonetheless, some concerns about the methodology were raised, especially statistical analysis to support the precision of estimates in the discussion of the results.

The main factor of novelty is the performance of a biosensor with immobilized colon cancer and healthy tissues, simulating the process whereby tissues (or cells) sense and identify carbon and nitrogen nutrients. Both, measurement and comparison of nutrition sensing kinetics between cancer cells and healthy cells, can be done with this apparatus.

We congratulate the authors for the excellent creativity and quality of the graphic images in Figures 1 and 7.

The specific comments are stated below:

#1_Line 267_Expression (2) shows a catalysis scheme but not the general hyperbolic equation to be fitted to the experimental data.  Please make clear which equation was used, identifying the variables and constants to be estimated.  It will be extremely important for non-enzymologists readers of Molecules journal.

#2_In the methodology session an appropriate statistical analysis is lacking. In which statistical package was the regression of the 11 experimental points (for each component variable: glucose, MSG, sodium lactate...) performed? Spss, Excel, R, JMP?

#3_Regarding the methodology in the kinetic modeling (graphical determination of parameters by linearization of the hyperbolic curve) we have some comments/concerns to announce.

#3.1. Could the authors clarify, please, the need to present a double reciprocal plotting of the data, previously fitted by non-linear regression which gives better precision of parameters than any linearization of the hyperbole?

#3.2. To be noticed that these hyperbole linearizations (Michaelis Menten-type) have been heavily criticized for decades, to name a few:

 I) "...KM estimates made using double-reciprocal plots and the MM equation were consistently inferior to estimates made with nonlinear least-square fitting methods" [Schnell, S. and Maini, P.K. (2003) A Century of enzyme Kinetics: Reliability of the KM and Vmax Estimates. Comments on Theoretical Biology, 8, 169-187. http://dx.doi.org/10.1080/08948550302453].

II) "Today, there is no reason for fitting data using either linear transformation of the Michaelis–Menten equation in analyzing the concentration dependence of the initial velocity" in [ Johnson 2013 A century of enzyme kinetic analysis, 1913 to 2013, FEBS Lett. 2013 Sep 2;587(17):2753-66. doi: 10.1016/j.febslet.2013.07.012.]

The authors are encouraged to focus on non-linear regression and to verify the precision of estimates for predictive models as suggested in comment #4.

#4_ In the Material and Methods, we didn’t find any Statistical analysis methodology to support the estimates of Tables 1-4. The preferred expression for an estimate and its precision is the mean and the 95% confidence interval (the range of values about 2 SEMs above and below the mean) [“Standard deviations and standard errors”_ DOI: 10.1136/bmj.331.7521.903]. To be noticed that “mean±SD” and “mean±SEM” are not the same. Please provide the precision of estimates and the residual graph (residuals vs predicted values) for the predictive models. In our opinion, the precision of predictive model estimations is vital for healthy vs non-healthy tissue comparisons in this study.

Reviewer 3 Report

The present paper deals Comparative study on the sensing kinetics of carbon and nitro- 2

gen nutrients in cancer tissues and normal tissues based electrochemical biosensors. Although the paper is interesting from a technical point of view, I accept it for possible publication in molecules in its present form. A minor correction is needed and the authors should consider all the following remarks and bring convenient answers to my questions as below:

Comment 1:

The protocol of preparation of biosensor described in this work is not clear. I suggest that the author add a reference

Comment 2:

Some grammar and spelling errors in the text should be avoided

Comment 3:

 I suggest the author give a more detail explanation in the introduction

Comment 4:

All figures should be improved

Comment 5:

After preparation the electrode you check the stability after a period of 15 days and 1 month.

Author Response

Q1: The protocol of preparation of biosensor described in this work is not clear. I suggest that the author add a reference.

A: Thank you for your good advice. References 21-25 have been added.

Q2: Some grammar and spelling errors in the text should be avoided.

A: Thanks for your kind suggestion. With the help of the native English speaker, we have revised the whole manuscript carefully. All revised words were shown in red color.

Q3: I suggest the author give a more detail explanation in the introduction.

A: Thanks for your constructive suggestion. The explanation of Figure 1 and the working principle of the sensor have been added in the introduction.

Q4: All figures should be improved.

A: Thanks reviewer’s good suggestion. All figures have been improved.

Q5: After preparation the electrode you check the stability after a period of 15 days and 1 month.

A: Thanks reviewer. Stability measurements for the sensors have been added in section 3.6.

Round 2

Reviewer 2 Report

We appreciate the author´s efforts in answering our questions. 

Scientific soundness has been improved.

The methodology was clarified and described more rigorously.